# A Methodology for Harmonizing Safety and Health Scales in Occupational Risk Assessment

**DOI:** 10.3390/ijerph18094849

**Published:** 2021-05-01

**Authors:** Zuzhen Ji, Dirk Pons, John Pearse

**Affiliations:** Department of Mechanical Engineering, University of Canterbury, Christchurch, Canterbury 8140, New Zealand; zuzhen.ji@pg.canterbury.ac.nz (Z.J.); john.pearse@canterbury.ac.nz (J.P.)

**Keywords:** risk matrix, health and safety risk, quality of life, World Health Organization disability assessment schedule, diminished quality of life, risk management

## Abstract

Successful implementation of Health and Safety (H&S) systems requires an effective mechanism to assess risk. Existing methods focus primarily on measuring the safety aspect; the risk of an accident is determined based on the product of severity of consequence and likelihood of the incident arising. The health component, i.e., chronic harm, is more difficult to assess. Partially, this is due to both consequences and the likelihood of health issues, which may be indeterminate. There is a need to develop a quantitative risk measurement for H&S risk management and with better representation for chronic health issues. The present paper has approached this from a different direction, by adopting a public health perspective of quality of life. We have then changed the risk assessment process to accommodate this. This was then applied to a case study. The case study showed that merely including the chronic harm scales appeared to be sufficient to elicit a more detailed consideration of hazards for chronic harm. This suggests that people are not insensitive to chronic harm hazards, but benefit from having a framework in which to communicate them. A method has been devised to harmonize safety and harm risk assessments. The result was a comprehensive risk assessment method with consideration of safety accidents and chronic health issues. This has the potential to benefit industry by making chronic harm more visible and hence more preventable.

## 1. Introduction

### 1.1. Context

Manufacturing is a potentially hazardous activity and is frequently the origin of occupational health and safety (H&S) issues. Conventional risk management methods, such as the standard ISO31000 [1], have been developed to measure risk. This process includes risks more generally, including opportunities, but in the H&S application the focus is on threats that result in human harm. Other methodologies with a strong safety emphasis are fault tree analysis (FTA) [2], failure mode and effects analysis (FMEA) [3], Bowtie [4], and hazard and operability study (HAZOP) [5], among others. Applications can be found in construction [6,7], manufacturing [8], chemical [9,10], and logistic industries [11].

The conventional risk assessment methods are mainly focused on safety incidents that would immediately cause harm to people, i.e., acute injury effects, as opposed to long-term (or chronic) health issues [12]. This is deeply problematic because recent policy developments in H&S have made it necessary for organizations to consider long-term health hazards, although there is no coherent methodology for performing due diligence to the legislative requirements. Consequently, organizations put themselves at risk of adverse legal consequences. The specific jurisdiction under examination is New Zealand. This country recently radically redeveloped its public safety legislation, taking into account best policy practice from around the world. The moral purposefulness of this radical transformation arose from the catastrophic failure of the Pike River mine (29 deaths) in 2010. It became apparent that existing legislation provided perverse incentives for organizations to underperform in H&S. In this case, the mining industry maintained that it was a self-regulating industry that only required light-touch legislation. There were many deficiencies at Pike River mine, including the gross inattention to hazard notifications by miners, the lack of a systematic and effective risk management system, the absence of adequate safe systems of work, the perverse incentives from management that caused workers to prioritize productivity over safety, and the potentially deliberate strategy of senior management to decrease culpability by keeping themselves uninformed of operation risk [13]. Consequently, the new legislation of New Zealand [14] specifically addressed many of these deficiencies, and in the process radically reshaped the social license to operate for organizations. A detailed review of the Act cannot be provided here, but for a short summary, see [15].

While other countries have not yet experienced the policy transformation that swept New Zealand in the 2015 H&S legislation, there can be no doubt that countries learn from each other; therefore, that similar policies may arise in other countries. There are still many nations that take the policy position of regarding H&S as primarily the prevention of accidents. Consequently, their risk assessments are primarily focused on the adverse outcomes that can be anticipated as the direct consequence of an accident—this was also the case in New Zealand. As a consequence, new methods, such as that described here, have to be developed to help meet those new legislative responsibilities.

The central purpose of this paper is to help overcome a major deficiency in industrial risk assessment, namely, the lack of any way to include long-term health outcomes. This work develops a new quantitative measurement to compute H&S risk, based on the integration of conventional risk assessment processes with the World Health Organization Disability Assessment Schedule Score (WHODAS), and the Diminished Quality of Life (DQL) method [12].

### 1.2. Difficulty of Predicting Long-Term Health Outcomes in Risk Assessments

Long-term health effects have been persistently difficult to include in H&S assessments, for several reasons. Fundamentally, the problem is that the etiology of later impairment is difficult to predict at the time of exposure. This can be because the consequences are not immediately apparent, or they accumulate in unpredictable ways, or there are threshold effects before impairment manifests. In addition, there may be age-related degeneration which may be exacerbated or triggered by occupational exposure, e.g., musculoskeletal pain and disorders due to persistent heavy load-bearing operations [16]. Each type of impairment has a different etiology—many are caused by cumulative exposure [17], such as hearing loss due to prior exposure to machinery noise [18,19]. However, there is limited methodological support for anticipating the long-term health issues when harm occurs by cumulative exposure or some period of time after exposure [12,20]. Furthermore, some impairments may be affected by multiple causes [12,21]. There is also a great deal of personal variability in the extent of impairment following exposure [19,22]. Hence, there is individual variability which further complicates the ability to generalize safety risks.

All of this makes long-term health effects extraordinarily difficult to predict in general. While prediction may be feasible in specialized occupational areas, such as exposure to ionizing radiation, this is not possible for industry in general. Consequently, it is difficult to attribute a health outcome to a specific industrial work activity. It is often impossible to retrospectively attribute impairment causation to specific exposures or associated industrial work, especially when workers undertake different tasks for different employers over their careers. Many health-related hazards are not sufficiently understood to permit precise quantification of exposure thresholds and consequences. Hence, long-term health aspects may be omitted from the risk assessment for the task, which instead focusses on the more immediate harm outcomes such as accident scenarios.

While it might theoretically be possible to quantify the probabilities of long-term harm effects, via correlation and conditional factors, the necessary level of data collection simply does not exist to sustain such an analysis. Hence, from the industrial perspective, it is important to be able to use the limited information available to produce some type of defendable analysis that can improve the situation for the benefit of workers.

There is a need to develop risk assessment methodologies that are better able to represent the chronic harm component alongside the immediate accident type events.

## 2. Research Approach

### 2.1. Research Purpose

The primary purpose of this work was to develop a risk assessment method for managing health and safety risks. Particular attributes are needed to address chronic health in the process, and to develop a more objective scale for such harm. The field of work under examination is the manufacturing industry.

### 2.2. Selection of Quality of Life as an Output Metric

The first part of the approach was to select a metric for health outcomes. As described above, it is impractical to quantify the probabilities of many outcomes with any precision. Hence, it was decided to take a more holistic approach, by focusing on the overall effects of occupational exposure rather than the specific causes.

Attempts to address this problem in the area of disease progression include quality-adjusted life year (QALY) [23,24] and Disability-Adjusted Life Year (DALY) [25]. These are widely applied in public health. However, these methodologies have drawbacks when applied to occupation health and safety. This is because QALY and DALY analyze the way diseases (such as cancer) affect a person’s health, using subjective estimation to determine the lost duration of life, sometimes with a quality adjustment. However, they do not include the full quality metrics of the more generalized quality of life scores. Furthermore, many occupational health and safety issues are chronic and not fatal in themselves, for example, hearing loss. In addition, these methods have generally not progressed to integration with a risk assessment, but rather are specialized models that address only one or a few health conditions.

The concept of quality of life has become important in medicine and public health, based on the realization that medical and surgical interventions can themselves cause loss of quality of life even if they prolong life (e.g., chemotherapy cancer treatments for older people). Numerous quality of life scores have been and continue to be developed, to help inform the decision as to whether treating the condition has advantages compared to living with it. One of the earlier and most influential scores is the WHODAS [26,27]. It uses multiple questions, on 0–4 scales, to measure the difficulty in self-care and household activities. It is a generalized metric, rather than being focused on one particular disease or condition, as the later scales tend to be.

Thus, quality of life has the potential to measure the severity of occupational health consequences. Instead of trying to score chronic harm risks with the conventional consequence and likelihood methods, or the adjusted mortality approach, a validated scale of quality of life may be used. In practice, there are a number of significant difficulties that need to be overcome to implement this. The first problem is that a mechanism needs to be found to relate the task facing the worker now, to the effects on future quality of life. This first problem already has a conceptual solution, in that the WHODAS has been adapted for measuring occupational health and safety, in the DQL method [28,29]. DQL has been developed as a quantitative method to measure the occupational health and safety risk at workplaces via integrating conventional risk assessment method and WHODAS [28,29]. However, it does this somewhat in isolation to the process for assessing accident risk. Hence, the second problem with the quality of life approach is the need to integrate it with the other types of assessment, especially of accident safety. This integration is addressed in the current paper.

### 2.3. Methodology for H&S Risk Harmonization

This approach develops a quantitative likelihood scale and consequence for H&S risk computation. The overall structure of the method is shown in Figure 1. Having adopted a method of risk determination for health in the form of the DQL [12,28], the next was to devise a method to integrate DQL and conventional safety risk assessment.

Underpinning this was the assumption that the health and safety aspects ought to be somehow complementary at a deeper, undiscovered level. Hence, a holistic mechanism was sought. We propose that DQL is not oppositional to the conventional safety risk assessment, but rather complements it, as shown in Figure 2. The solution adopted here was to devise new dual-consequence and likelihood scales.

The decision-thresholds for risk management were then identified. There is no standard of risk threshold in the conventional methods. Instead, most organizations develop their own risk tolerances, typically expressed as three or more colors in the risk register. Solving this problem becomes more complex when long-term health is added. The solution approach taken in the present paper was to adapt recent standardization of the accident thresholds [15]. These risk thresholds were determined by the application of a non-linear consequence scale. The consequence scale was developed to be consistent with the H&S legislation of New Zealand, because lack of such consistency is another issue of the application of the conventional safety risk assessment.

Finally, the harmonization method was applied to one of the production units (pie making) at a food production company. The pie making process is a partly manual and partly semi-automatic process. Operations including automatic pie forming, manual potato topping, oven cooking and semi-automatic packing. The primary hazards associated with the operation including lifting heavy loads, exposure to noise, and inhalation of flour dust. Ethics approval was obtained from the University of Canterbury (HEC 2019/28/LR-PS) for the data collection and interviews. The risk parameters such as frequency of an incident arising, likelihood of the harm occurrence, and severity of harm were determined from the health and safety representative from the company. We then determined the scale for consequence based on [12], where WHODAS was used to evaluate the level of harm and assist the development of a consistent consequence scale.

## 3. Results

### 3.1. Harmonized Likelihood Scale

Likelihood was determined by two parameters, namely, the product of the frequency of the incident arising, and the probability of people being harmed. In the original DQL formulation, each of these scales is a 1–6 ordinal range: The commonly used ordered scale is 6: Almost certain 100%, 5: Likely 60%, 4: Possible 40%, 3: Unlikely 20%, 2: Rare 10%, and 1: Almost Incredible 1%.

The DQL instrument spreadsheet represents the likelihood algorithm, with columns D and F being the input likelihood, and H being the product, as shown in Figure 3. Up to this point, nothing has been changed in the DQL calculation as originally proposed, so this itself is not novel.

The likelihood of harm based on the existing likelihood scale [12,30] was then re-evaluated. The scales ‘almost certain’, etc., are common ones used by industry, at least in New Zealand. This is because the ISO31000 risk management standard originated as a joint standard development in New Zealand and Australia and had a long life as standard AS/NZS 4360 and its accompanying explanatory handbook SAA/SNZ HB436. The standard subsequently became ISO31000, and the handbook continued as an Australia–New Zealand publication [30] that was current at the time of writing. Therefore, there is a large body of existing practitioner knowledge and skill regarding conducting risk assessments. For this reason, we started with the well-established 1 ‘Almost impossible’ to 6 ‘Almost certain’ scale for likelihood. However, we found that it showed a lack of clear progression in ranked order, as shown in Figure 4. Furthermore, it was not conducive to a wider integration with harm. Hence, the scale needed to be modified, as shown below, although we retained the six steps for practitioner familiarity.

We then developed a new likelihood scale based on the following principles:Needs to be consistent by showing a defendable progression;Needs to provide ordinal numbers for a descriptive scale;Needs to accommodate a high probability of many events of minor consequence, and low probability of events with extreme consequences;Needs to use simple numbers (easy to compute)—this is so the new method can easily be used by industry practitioners in the field on paper, where most risk assessments are conducted.

Consequently, we propose that a scale of 0.1 to 10 is better, because the reduction in harm is slower than the previous scale. This means that more importance is given to hazards with low WHODAS but occurring more frequently. The new curve was determined, and is shown in Figure 5. This distributed more points into the lower part of the likelihood scale, while still preserving the convenience of whole numbers. We wish to emphasize that this scale was constructed for the purposes of integration with the risk assessment. Whether or not people actually interpret these likelihoods with the words is not the purpose of the current study, and is not relevant, because the scale of Table 1 can be provided to them beforehand to frame their interpretation. The descriptor text was adapted from [30] and includes a time dimension.

### 3.2. Harmonizing the Scale of Consequence

The consequence scale needed to be harmonized between accident and long-term health outcomes. The challenge here was to devise one scale that accommodated both health (measured by the WHODAS score), and the conventional safety assessment (typically 1 to 5 or something similar). The solution we selected was to adapt the latter to the WHODAS scale.

The natural range of WHODAS scores is from 0 to 100. In order to achieve the integration between this health scale and the safety scale, we recalibrated the safety scale. Before doing so, we enlarged the WHODAS scale to a maximum of 500 to accommodate more serious situations where numerous people could be affected. This was justifiable because the WHODAS is focused on measuring health outcomes for one individual person. We also categorized the WHODAS score into levels, and devised a harm description for each, e.g., 20 < WHODAS < 30 corresponds to ‘Serious harm, e.g., amputation’ (see Table 2).

Next, it was necessary to align the subcategories for the health (WHODAS) scale with the conventional safety scale developed for the New Zealand HSAW [28]. The latter scale may be stretched to match the upper point (500) of the health scale. Most conventional risk scales in use are linear ordinal scales, which have the limitation of representing serious outcomes (such as death) as only a few steps away from trivial outcomes. Hence, greater consideration must be given to the subcategories. The subcategories for the safety scales are the levels of harm per the relevant health and safety legislation. Typical categories are ‘incident’, ’minor harm’, ‘serious harm’ or other such progression. These are semantically different in each jurisdiction; nonetheless, there are common features, and a method is available to provide numerical scores to such subcategories [28].

There were several rounds of iteration between the reformatting of the WHODAS health scale, and a similar exercise for the safety scale. Harmonization was achieved by (a) adopting the same number of categories in each scale, and (b) aligning permanent debilitating harm with serious harm and assigning it with a score of WHODAS = 20. This gave two parallel scales, one for the immediate harm from an accident (safety), and the other for chronic harm (health). The results are shown in Figure 6.

### 3.3. The DQL Risk Matrix

Finally, a harmonized risk assessment matrix was developed, as shown in Figure 7. This used the new dual-consequence and likelihood scales shown in Figure 6. This is a further development of [28], with the addition of the scale for long-term health, and a recalibration of the consequence and likelihood scales.

The numerical values shown in the body of the matrix are the product of consequence and likelihood. They represent the risk.

The intended use in a practical setting would be to consider each threat from both short- and long-term health perspectives, by entering the table on the relevant consequence scale. The scale accommodates both safety and health perspectives.

We suggest that it would be beneficial for organizations to use a consistent set of consequence and likelihood scales such as this—rather than the ad hoc constructs currently in use—for better benchmarking between organizations and industrial sectors.

### 3.4. Risk Appetite and Response Scale

The colors in the risk matrix represent the organizational risk appetite. This specifies the level of risk that an organization is willing to take, and who must be involved in the decision-making. It is a form of delegation of responsibility for risk management to its various operational levels. Risk assessments are performed by different people in an organization; therefore, the risk appetite sets a common understanding of where the thresholds are and with whom to communicate. The thresholds are represented as color codes in the risk matrix. They also correspond to instructions on how people are expected to respond. Current practice is for organizations to set their own risk appetites, and hence there is a large degree of variability in the outcomes. This is exacerbated by organizations using different consequence and likelihood scales, hence having different numerical values of the risk product.

Generalized consequence and likelihood scales have been demonstrated above; therefore, it is now possible to also make some suggestions for a generic set of risk appetites. In doing so, we adapt the approach of [28]. This assumes an organizational structure of workers/operators, team leaders, technical managers, executives (CEO), and board (governance). Most organizations can be approximated by these structural categories, although some have more or less. This provides five levels within the organization, which we interpret as a call for five risk thresholds and hence the same number of colors.

It then becomes necessary to decide how to map these five categories to numerical risks. In doing so, we have given regard to the H&S legislation for New Zealand, where there is a requirement that executives (called ‘officers’ in the Act) keep themselves informed of H&S risks within the organization. The logical test case we applied was that executives should be informed of any threat where it is likely that serious harm might occur, or executives could be personally liable. This position is subjectively identified on the scales as having a risk of 180. This becomes the lower point or threshold at which subordinates need to escalate the matter to the executive. In practice, we set this threshold conservatively and slightly lower, at 120, to avoid having gaps in the response scale.

The resulting response scale is shown in Table 3, with the colors having been anticipated in Figure 8. This is a further development of [28], with the primary changes being the addition of the grey category (extremely urgent cessation of activities), inclusion of a new worker category, and the adjustment of the thresholds to accommodate the new risk scales. Note that the table is a suggested representation of decision thresholds, and individual organizations would need to adapt this for their own risk appetites.

### 3.5. Application of the Harmonized H&S Risk Assessment Method to Case Study

Finally, we illustrate the application of this new method to a food production case study. The operations under review are associated with pie production. The operations are conducted with uncomfortable thermal environments (e.g., working near hot ovens and inside chillers) and repetitive movement (food preparation and packing processes). These hazards can further result in long-term health issues, for example, musculoskeletal injuries. The H&S risks were measured using both conventional and the new method. The severity of harm and likelihood were determined through onsite observation, and confirmed by the operations manager and H&S representative. The results were then discussed with the general manager, production team leaders, and H&S representative for validating the accuracy and reliability of the results.

Take, for example, flour dust. The potential issues here are respiratory inflammation, e.g., asthma, sinusitis, etc. These can develop longer term adverse outcomes. The purpose of the dual likelihoods is to determine (a) the likelihood that there will be air-borne flour in the bakery environment with whatever protections are currently in place (risk assessment is always conducted assuming the current state of the industrial plant), and (b) the likelihood that this will result in adverse consequences (long-term harm outcomes). The magnitude of those consequences is also determined. As with any risk assessment, this evaluation is performed by the workers, team-leaders, and H&S representatives, and takes into account what they know about the hazard, and their own experience.

The results of the case study based on using the conventional safety assessment method [15] are shown in Table 4, and for the new method in Table 5. Although the values of risk parameters are changed—because of the new risk scales—the required treatment outputs are similar.

Compared to the conventional risk assessment, the new harmonized method elicits more hazards. Notable outcomes are that the new method anticipates modes of injury that were not apparent in the conventional risk assessment. Consequently, the proposed new approach using harmonized safety and health scales is considered to be successful at (a) making health hazards more explicit in the risk identification process, and (b) providing a mechanism to manage safety and health risks in a single framework.

## 4. Discussion

### 4.1. Interpretation

Compared with conventional risk assessments, the harmonized health and safety risk assessment encourages a greater emphasis on the health aspect. Consequently, long-term health issues such as dust exposure, noise, repetitive movements, trips, etc., are more likely to be identified in the initial hazard identification process, as the case study shows. It appears that the explicit inclusion of a chronic harm scale heightens the cognitive awareness of such hazards. The conventional safety scale predisposes the analysis towards accident scenarios, and by not having an easy mechanism to include chronic health it may cause these outcomes to be overlooked.

### 4.2. Implications for Practitioners

Practitioners will be familiar with hearing loss and other long-term health outcomes of their place of work. In applying the method proposed here, practitioners are recommended to consider any widely accepted thresholds and heuristics for safe levels and durations for the various occupational hazards. This type of information is commonly available from the national H&S regulator or safety institutions. While the information is not quantitative or algorithmic, we believe it is nonetheless sufficient for an industry person to determine whether or not the work activities under consideration (including any existing treatments) are in accordance with the guidelines given in the practitioner literature. If so, the risk can be evaluated as low. If not, then the risk is elevated, and may pass the threshold on the risk appetite to require better treatments.

The current paper addresses the analysis tasks at the front end of the H&S process. This is only one part of a wider H&S management system. The main components to such a system may be categorized into risk assessment, the development of procedures, commitment from managers and workers, and monitoring and ongoing improvement. In some industries, the monitoring of worker health (e.g., hearing, radiation exposure) may be a large part of the safety management systems. Documentation is essential, to ensure that opportunities for improvement that are identified (by risk assessments or incident reporting) are not lost. All safety systems include an element of ongoing improvement. Commonly, this uses the Plan–Do–Check–Act (PDCA) cycle from the continuous improvement methodology. Another improvement approach is provided by kaizen, which is characterized by small group interactions and creative problem-solving, and can also use PDCA. By whichever method, improvement plans are devised and enacted. Again, because the risk assessments provide a starting point for analysis of the hazards across multiple different work processes, this is another benefit to having the assessments on common scales. Hence, while the method developed in this paper offers a way to facilitate the early hazard analyses, it is not recommended to be used in isolation but rather adapted for inclusion in a more holistic H&S management system.

### 4.3. Limitations and Further Research Opportunities

All hazard assessment methods, including this one, rely on a subjective assessment of the likelihood of exposure. This is not necessarily a problem, because the purpose of the assessment is to (a) expose risks, especially those that are hidden to the organization, and (b) provide a means to rationally order the risks for treatment. That ordering process needs to be performed in a defendable way, i.e., there is a need to show due diligence to other stakeholders (more senior managers, legislative duties, etc.). Hence, a comprehensive inclusion of hazards is more important than the exact numbers that come out of the risk assessment process. What is valuable is internal consistency within the organization, so that the multiple risks assessments are conducted similarly. The reason that this is important is that resources for treatment are finite, therefore the results from multiple assessments need to be compatible in cross section (with each other) and longitudinally (over time, to check for trends). To help with this, we propose that the determination of harm should be undertaken by a multidisciplinary team, so that multiple perspectives can be applied; for example, a team comprising health and safety representatives, workers, engineers, and operation managers. There is another reason for including others in the risk assessment process, which is that it helps develop a common understanding of risk, i.e., a communication benefit.

We admit that with a comprehensive examination using quantitative method to determine OHS risk may be helpful, and this would also further assist H&S representatives or managers to better understand the hazard exposure, and severity of consequences. However, this would require extensive work from the start—“engineering design”—to the end—“production operations”. This is challenging, because not all these engineering processes examine OHS risk quantitatively, but rather use subjective analysis such as FTA and Bowtie methods. Additionally, some new production systems or devices lack such OHS risk data; hence, specialists usually determined the risk using subjective analysis, which is based on their knowledge and experience. Nevertheless, we propose that there is a possible solution which combines both quantitative assessments and subjective assessments in risk determination. This idea has been described in Figure 8. We propose that the new risk determination process should include two routes, one for the initial risk determination, because in this circumstance, when there is a lack of OHS data, it is difficult to apply quantitative assessments and hence subjective assessments are more useful and applied first. Then, from time to time, when the data of OHS have been recorded sufficiently, the specialists could re-evaluate the risk of the production system; hence, quantitative assessments are then applied. We also propose that this risk determination process is not a one-time decision, but a continues evaluation process.

Loops of causality were not considered in this work. For example, many operations in manufacturing can cause noise, and this cannot only cause hearing loss, but may further affect a person’s mood and fatigue. Another limitation is that mental health was not considered. The WHODAS does not measure mental health issues. An opportunity for future extension may be to apply a third scale for mental health.

## 5. Conclusions

A method has been developed to include chronic harm in the risk assessment process. This provides a solution to a complex, long-standing problem. Previous approaches have grappled with two problems: the difficulty of devising a scale to measure chronic harm, and the difficulty of combining short- and long-term harm in one assessment method. They have been underpinned by an engineering rationality of seeking to quantify chronic harm—those efforts have not been productive because harm is too complex to be represented as such. The present paper has approached this from a different direction, by adopting a public health perspective of quality of life. We have then changed the risk assessment process, primarily the scales, to accommodate this (rather than the other way round). The case study showed that merely including the chronic harm scales appears to be sufficient to elicit a more detailed consideration of hazards for chronic harm. This suggests that people are not insensitive to chronic harm hazards, but benefit from having a framework in which to communicate them.

The novel intellectual contribution here was developing a method to harmonize safety and harm risk assessments. The result is a comprehensive risk assessment method with consideration of safety accidents and chronic health issues. This has the potential to benefit industry by making chronic harm more visible and hence more preventable.

## Figures and Tables

**Figure 1 ijerph-18-04849-f001:**
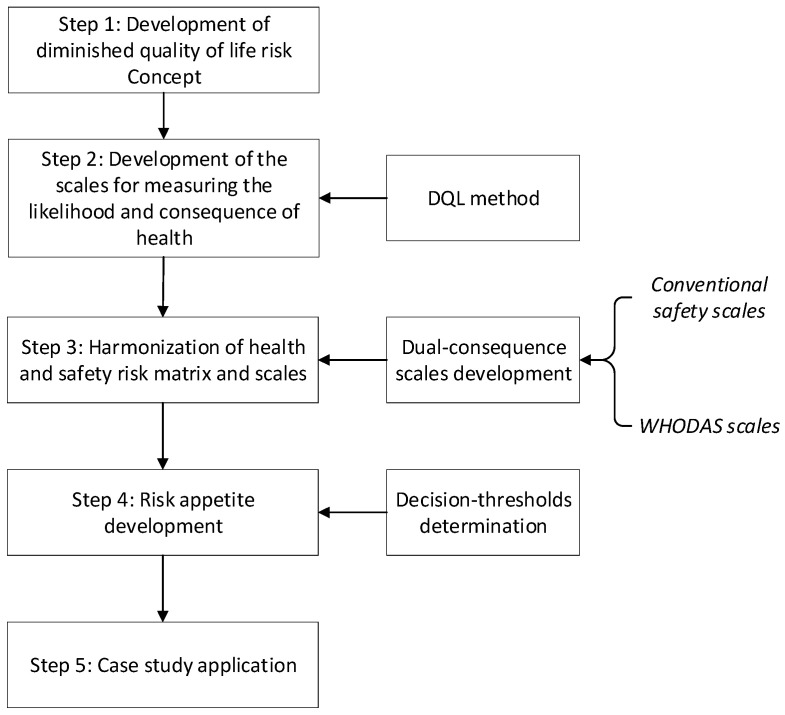
Research workflow of the development of H&S risk harmonization.

**Figure 2 ijerph-18-04849-f002:**
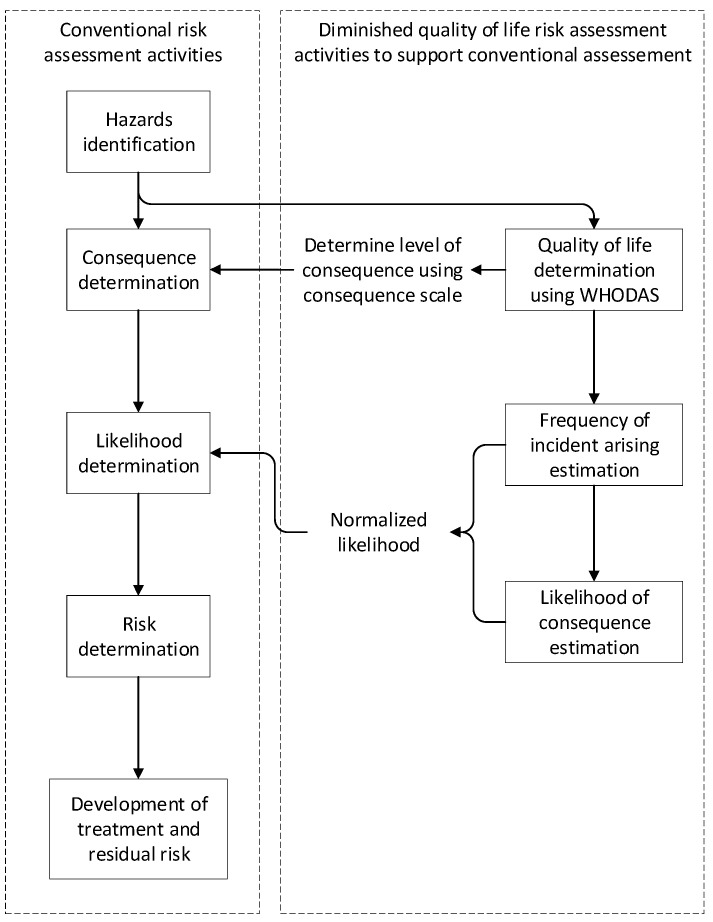
The integration of conventional risk assessment and DQL method.

**Figure 3 ijerph-18-04849-f003:**
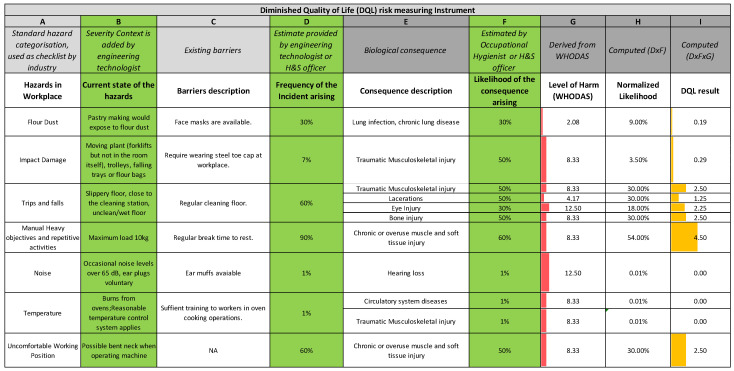
Revised DQL instrument.

**Figure 4 ijerph-18-04849-f004:**
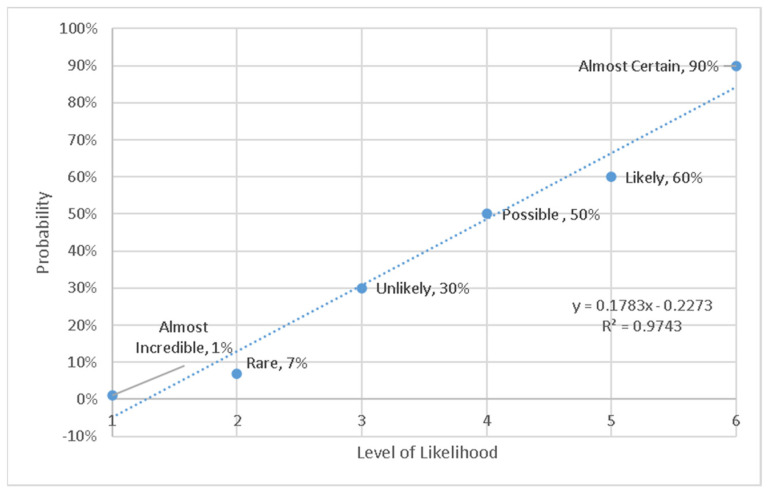
Original likelihood scale [12] (scale from Almost Incredible “1” to Almost Certain “6”).

**Figure 5 ijerph-18-04849-f005:**
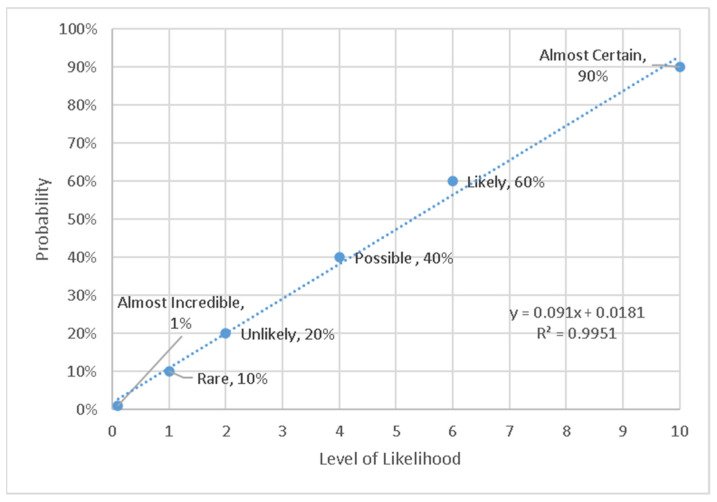
Revised likelihood scale (scale from 0.1 to 10).

**Figure 6 ijerph-18-04849-f006:**
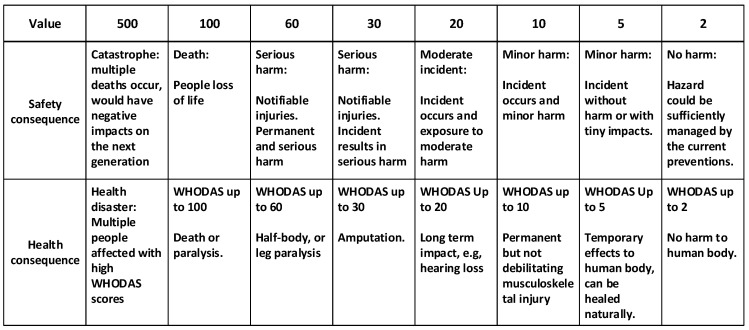
Severity of consequence description.

**Figure 7 ijerph-18-04849-f007:**
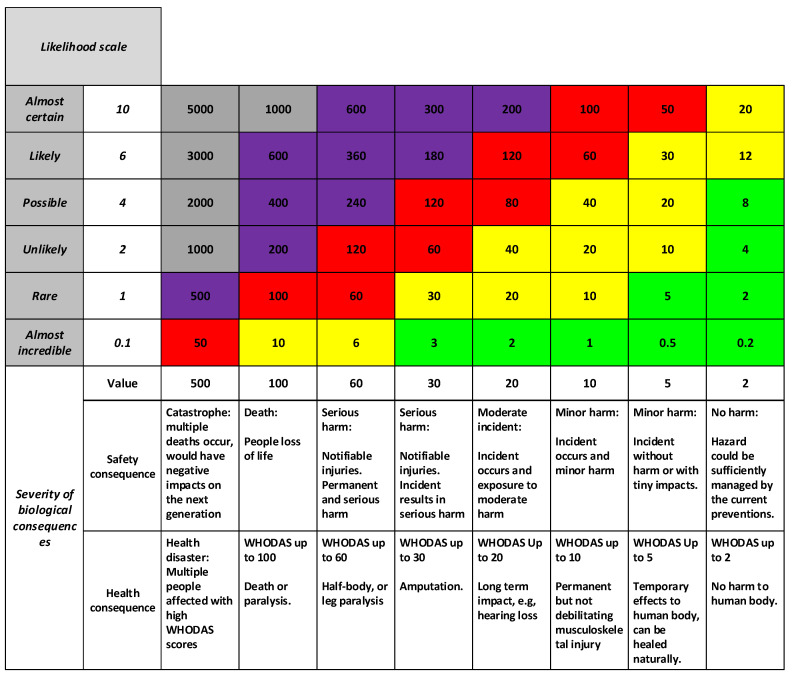
Risk matrix with harmonized consequence scale for safety and health outcomes, and the new likelihood scale.

**Figure 8 ijerph-18-04849-f008:**
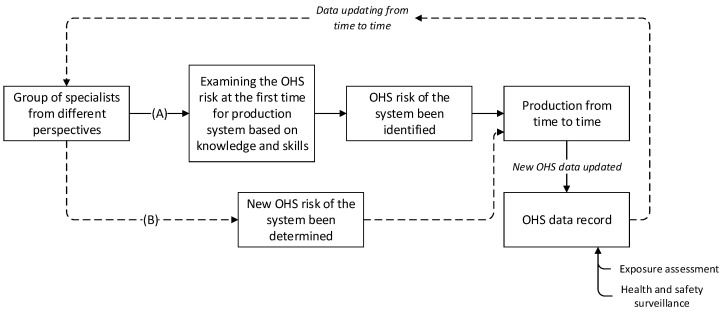
Workflow of the integration method.

**Table 1 ijerph-18-04849-t001:** Revised likelihood scale.

Likelihood Description	Likelihood Scale	Descriptor (Adapted from [30])
Almost Certain	10	Annual occurrence in this situation
Likely	6	Has occurred several times in your career
Possible	4	Might occur once in your career
Unlikely	2	Event does occur somewhere from time to time
Rare	1	Heard of something like this happening elsewhere
Almost Incredible	0.1	Theoretically possible but not expected to occur

**Table 2 ijerph-18-04849-t002:** WHODAS categories.

Level of WHODAS Score	Harm Description
100 < WHODAS < 500	Large group of people been affected, a health disaster, could have negative impact on next generations
60 < WHODAS < 100	Death or very serious harm
30 < WHODAS < 60	Serious harm, e.g., half-body paralysis.
20 < WHODAS < 30	Serious harm, e.g., amputation.
10 < WHODAS < 20	Moderate but permanent harm
5 < WHODAS < 10	Minor harm, permanent but not debilitating
2 < WHODAS < 5	Minor harm, temporary effects to human body, easy to heal
0 < WHODAS < 2	No harm to human body.

**Table 3 ijerph-18-04849-t003:** Risk appetite response scale.

DQL Score (C × L)	Severity of Harm (Color in Risk Matrix, See Figure 7)	Description of Treatment	Actions	Authority for Continued Operation	Reporting
DQL > 1000	Grey	Cessation.	Immediate intermission must be undertaken. Ensure preventions and recoveries are adequate and can manage the risk in the future operations.	Board members	CEO must report and advise solutions to Board members under urgency.
120 < DQL < 1000	Purple	Unacceptable risk.	Cease operations immediately until risk has been minimized. Ensure preventions and recoveries are sufficient and it is possible to manage the risk in the future operations.	Board and CEO	CEO needs to report and advise solutions to the Board as soon as practicable.
60 < DQL < 120	Red	Urgent treatment.	Urgent treatment required. Operations proceed with caution and ongoing monitoring of risk.	Technical manager	Technical manager to advise CEO as soon as possible, and report regularly on status of the risk and its treatment.
10 < DQL < 60	Yellow	Consider treatment.	Implement treatment in a reasonable time period. Continue the operations with caution. Monitor the risk in case it becomes worse.	Team leader	Team leader to report regularly to Technical manager on the risk and the progress of the treatment plan.
0 < DQL < 10	Green	Not necessary to have special treatment.	No special treatment required. Continue the operations with ongoing monitoring of the efficacy of existing preventions.	Operators	Staff to report regularly to Team leader on the state of this risk.

**Table 4 ijerph-18-04849-t004:** Conventional safety risk assessment result.

Hazard Description	Level of Consequence (C)	Likelihood of the Issue Occurrence (L)	Risk (C × L)	Principle of Action
Repetitive activities	3	6	18	Urgent treatment needed.
Noise	3	3	9	Consider treatment
Electrocution	5	2	10	Consider treatment
Worker entrapped by operating conveyors	5	3	15	Consider treatment

**Table 5 ijerph-18-04849-t005:** Harmonizing risk assessment result for case study.

Hazard Description	Level of Consequence (C)	Normalized Likelihood (L)	DQL Score (C × L)	Principle of Action
Inhale flour	2	1	2	No further treatment required.
Electrocution	30	1	30	Consider treatment
Impact damage to human body	10	2	20	Consider treatment
Noise	10	2	20	Consider treatment
Repetitive operations	10	6	60	Urgent treatment needed
Uncomfortable working environment—temperature	10	0.1	1	No further treatment required.
Trip issue	10	2	20	Consider treatment
Uncomfortable working positions	10	2	20	Consider treatment
Trapped by equipment, e.g., conveyor	30	1	30	Consider treatment

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
