# Peer review of "A Methodology for Harmonizing Safety and Health Scales in Occupational Risk Assessment"

_ijerph, 2021, doi:10.3390/ijerph18094849_

Round 1
Reviewer 1 Report
The Authors have improved the manuscripts considerably. Hence, in this reviewer's opinion no further augmentations are needed and it can be considered for publication as it is.
Author Response
We thank you for this positive comments.
Reviewer 2 Report
Dear Editor,
my opinion was to reject the paper because in my field (occupational hygiene) the monitoring is considered as the gold standard. However, I can guess that in broader fields (HSE management) this would not apply.
I have not changed my mind and I think the manuscript level is scientifically low from an occupational hygiene point of view while I admit that its practical importance for consultants and professionals can be high.
So, I would recommend to reject or rather involve in the review process another expert in Occupational Safety or HSE management.
Best regards,
Author Response
We understand the reviewer’s concern, and to address it we have added text to make it clearer that the method only addresses one part of a broader H&S management system.
Please see details in additional text (highlighted) in Discussion.

Reviewer 3 Report
The manuscript was significantly improved and the authors answered the reviewer's doubts.
Author Response
We appreciate your positive comments.
Round 2
Reviewer 2 Report
I appreciated the acknowledgement that the paper topic deals with a part of a wider H&S assessment and management framework, as well as the effort to bring different disciplines together.
In my opinion, the proposed solution for combining subjective and quantitative assessments is also to be welcomed. The first are crucial to set up a HSE management system, while quantitative approaches should be used to test its efficiency/efficacy.
In short, I consider this revised version acceptable for publication.
I would take the occasion to suggest authors to include also health experts (occupational physicians and hygienists) within the multidisciplinary team and to use "exposure assessment" instead of "hazard exposure e.g. frequency of exposure" and "health and safety surveillance" instead of "severity of consequences" in Fig.8.
This manuscript is a resubmission of an earlier submission. The following is a list of the peer review reports and author responses from that submission.
Round 1
Reviewer 1 Report
The study introduces a very interesting methodology for the assessment of chronic harms in occupational risk assessment procedures, providing a novel tool for taking into account both safety and health perspectives.
The manuscript is well written and organized. However, it presents several critical flaws that need to be addressed.
First, in the introduction, the research motivations should be explained in a clearer manner showing why the proposed methodology is needed and how it can augment knowledge in the risk assessment literature. Actually, numerous methods for occupational risk assessment (ORA) have been proposed in the literature, and their analysis should be expanded to support research motivations both considering managerial, process oriented or qualitative perspectives (e.g. you might consider the following studies: http://dx.doi.org/10.1016/j.ssci.2017.03.021; https://doi.org/10.1080/10803548.2018.1483100; http://dx.doi.org/10.1016/j.ssci.2011.01.003).
The research approach should include (i.e. Section 3) should include a complete description of the proposed procedure. Accordingly, subsections 4.1-4.3 could be moved here, while Section 4 should illustrate the practical application of the method to a case study.
With reference to the latter, a comparison of results achieved by means of traditional risk assessment could help the reader in better understanding the value of the proposed methodology.
Also Figure 12 should be included in the description of the methodology.
The discussion of results should be expanded, illustrating more in detail how the proposed tool differs from other approaches presented in the literature, especially those based on the legislative framework for the evaluation of the risks (in terms of likelihood and consequences), as for example in https://doi.org/10.3390/ijerph16030310. Moreover, the sentence in lines 279-280 (page 11) needs to be explained, since it is unclear so far how the mentioned study [13] has implications with the current study.
Finally, the study limitations should be also mentioned, discussing whether there is a specific application field/context of the proposed tool, and its extendibility.
Additionally some minor concerns regard the followings:
- The abstract should follow the journal rules;
- Figure 8 could be divided into two different figures, since as it is, it results quite unclear how to use the matrices.
- Figure 9: this could be a table, as correctly mentioned in line 234. In any case a different formatting is needed to improve its quality.
Reviewer 2 Report
The manuscript is aimed at proposing a new methodology for a holistic and harmonized risk assessment of safety and health hazards in occupational settings.
The topic of this paper fits with one of the IJERPH aims and scopes (occupational hygiene), the manuscript is well organized, and the new methodology is clearly presented.
I am aware that this methodology may have important practical implications and could be very interesting for policy makers, but (I am sorry to say) I do not consider the manuscript suitable for publication, except after a substantial change of vision on what is essential for an objective health risk assessment.
Despite I can agree with the starting point, namely that traditional safety risk assessment methods are not appropriate to represent chronic H&S risks, I personally think that the proposed methodology is not enough scientifically sound and does not consider the scientific expertise accumulated in decades of studies, discussions and debates in the field of occupational hygiene.
As reported a few years ago by Hans Kromhout in his Editorial “Hygiene Without Numbers” (2016), a scientific health risk assessment, for preventing either acute or chronic effects, should unavoidably rely on a robust exposure assessment and not on the simplistic methods based on hazard/exposure/risk scales or bands that are increasingly applied by HSE managers and corporate consultants.
Health risks are already complex per se (e.g. identification of critical effects among all those possible, dose-threshold mechanisms of actions or not, role of individual susceptibility...) and therefore the risk assessment, which is a quantitative process by definition, should be founded on quantitative dose-response markers (industrial toxicology and occupational epidemiology) and on a solid exposure assessment (occupational hygiene) built on a representative number of exposure levels measured via environmental or biological monitoring or estimated through validated exposure models. Thus, without exposure numbers in our hands, we cannot develop meaningful interventions and confidently monitor progress in preventing hazards from becoming risks.
Reviewer 3 Report
The topic is interesting and under the journal scope. However, the reviewer has some concerns about the methodology applied. It is suggested several improvements:
- In the Keywords list, the authors used an acronym with no previous explanation (WHODAS and DQL). Please revise.
- In Lines 49-50, you mention ISO and Legal documents (European and New Zealand). Please clarify and explain why we mention these different documents.
- Line 51, the meaning of FTA, FMEA, HAZOP must be included.
- Line 57, the authors mentioned that many health consequences are caused by prolonged exposure, such as hearing loss. It is correct, but a more complete revision should be done, mentioning other studies and consequences (e.g. work-related musculoskeletal disorders).
- Line 93: The research purpose must be justified/supported by previous studies, highlighting the research gap and importance/motivation of the current study. However, it should be addressed t the last part of the Introduction.
- Line 96: The authors did not explain and justify the different methodological steps of their study. The description of the methodology applied must be significantly improved.
- E.g.: Line 99 - the risk assessment was developed for all workstations/tasks of the company? How many workers were involved?... Please, describe the application of the DQL and WHODAS, exemplifying the actual application. This should be linked with your specific study.
- Line 108: the decision-thresholds should be presented and clarified.
- Line 114: add the meaning of NZ.
- You presented results that you did not explain its obtaining. As aforementioned, the methodology should be more detailed and comprehensive.
- Line 209: The number of the subsection is missing.
- concept of "risk appetite" must be supported by bibliographic reference/s.
- Figure 9, 10, 11 are tables, not figures.
- Lines 268, 276 and 283: Numbering of subsections is missing;
- The discussion of results is too limited and the authors did not compare their results with previous studies.